# Human Papillomavirus-Related Non-Metastatic Oropharyngeal Carcinoma: Current Local Treatment Options and Future Perspectives

**DOI:** 10.3390/cancers14215385

**Published:** 2022-11-01

**Authors:** Michaela Svajdova, Pavol Dubinsky, Tomas Kazda, Branislav Jeremic

**Affiliations:** 1Department of Radiation and Clinical Oncology, General Hospital Rimavska Sobota, 979 01 Rimavska Sobota, Slovakia; 2Department of Radiation Oncology, Faculty of Medicine, Masaryk University, 625 00 Brno, Czech Republic; 3Department of Radiation Oncology, Masaryk Memorial Cancer Institute, 602 00 Brno, Czech Republic; 4Department of Radiation Oncology, East Slovakia Oncology Institute, 040 01 Kosice, Slovakia; 5Faculty of Health, Catholic University Ruzomberok, 034 01 Ruzomberok, Slovakia; 6School of Medicine, University of Kragujevac, 340 00 Kragujevac, Serbia

**Keywords:** oropharyngeal cancer, human papillomavirus, deintensification, minimally-invasive surgery, radiation therapy, systemic therapy, de-escalation, quality of life

## Abstract

**Simple Summary:**

Current options for the curative treatment of the non-metastatic oropharyngeal carcinoma associated with human papillomavirus include minimally-invasive surgery or radiotherapy, which is combined with chemotherapy in many clinical situations. The aim of this review is to summarize the current treatment recommendations in the local therapy of human papillomavirus-associated oropharyngeal cancer, with respect to the latest published evidence in the field. Future perspectives and next-generation research pathways, including treatment deintensification methods, are thoroughly discussed.

**Abstract:**

Over the last two decades, human papillomavirus (HPV) has caused a new pandemic of cancer in many urban areas across the world. The new entity, HPV-associated oropharyngeal squamous cell carcinoma (OPSCC), has been at the center of scientific attention ever since, not only due to its distinct biological behavior, but also because of its significantly better prognosis than observed in its HPV-negative counterpart. The very good treatment outcomes of the disease after primary therapy (minimally-invasive surgery, radiation therapy with or without chemotherapy) resulted in the creation of a separate staging system, reflecting this excellent prognosis. A substantial proportion of newly diagnosed HPV-driven OPSCC is diagnosed in stage I or II, where long-term survival is observed worldwide. Deintensification of the primary therapeutic methods, aiming at a reduction of long-term toxicity in survivors, has emerged, and the quality of life of the patient after treatment has become a key-point in many clinical trials. Current treatment recommendations for the treatment of HPV-driven OPSCC do not differ significantly from HPV-negative OPSCC; however, the results of randomized trials are eagerly awaited and deemed necessary, in order to include deintensification into standard clinical practice.

## 1. Introduction

Head and neck squamous cell carcinoma (HNSCC) is a widespread type of cancer arising from the mucosa of the organs of the head and neck region. Oropharyngeal squamous cell carcinoma (OPSCC) comprises the anatomical regions of the palatine and lingual tonsils, base of the tongue, soft palate, uvula, and the lateral and posterior wall of the oropharynx. The aim of this comprehensive review is to summarize the current treatment recommendations in the local therapy of human papillomavirus-associated OPSCC, with respect to the latest published evidence in the field. Future perspectives and next-generation research pathways, including treatment deintensification methods, are thoroughly discussed.

### 1.1. Epidemiology 

Head and neck squamous cell carcinoma has been traditionally associated with alcohol consumption and smoking; however, a reduction in tobacco use in most high-income countries over the past two decades has led to a decline in its incidence [1]. At the same time, carcinogenic human papillomavirus (HPV) has emerged as an important risk factor in the pathogenesis of OPSCC. Its incidence is rapidly increasing in high-income countries across the world [2], making it the most common HPV-related malignancy, which has now surpassed even cervical cancer [3]. The global percentage of HPV-related OPSCC was estimated to be 33% in 2021, with a remarkable variability in the geographical distribution of the disease [4]. The most significant reported rise in its incidence was observed in the United States [2], Germany [5], Netherlands [6], Scandinavia [7,8], Italy [9], Brazil [10], and Thailand [11]. Regarding the anatomical region, the highest rate of HPV positivity (HPV+) is located in the palatine tonsils and the base of the tongue [4]. Oral HPV infection is transmitted either via oral sex, open-mouth kissing, or contact between the mouth and anorectal region, making HPV+ OPSCC the most rapidly spreading sexually-transmitted disease in the world [12]. Sexual behavior is a well-described risk factor for HPV+ OPSCC, with a strong association between the total number of sexual partners and the incidence of the disease [13]. However, HPV16 deoxyribonucleic acid (DNA) is only rarely detected in the oral rinses of long-term partners of HPV+ OPSCC patients, with a cumulative risk of active infection being approximately 4% [12]. Increased rates of HPV+ OPSCC are being observed in both younger and older adult patients, with most cases reported in white men *<*65 years of age [14]. 

### 1.2. Biology of HPV+ OPSCC

The molecular characteristics of HPV+ OPSCC differ significantly from HPV− OPSCC, making it a unique biological entity. The high-risk HPV16 subtype of the human papillomavirus is a causative agent in the majority of clinical cases. The function of the tumor suppressor p16 protein is to bind to the cyclin D1 cyclin-dependent kinase (CDK) 4/CDK6 complex and thus prevent Rb protein phosphorylation. Therefore, HPV+ OPSCCs typically present with overexpression of p16 protein, wild-type p53, and downregulation of pRb, whereas in a typical HPV− OPSCC, frequent p53 mutations are accompanied by increased levels of pRb and decreased levels of p16 [15]. Immunohistochemical staining for p16 is highly sensitive for HPV+ OPSCC and is routinely used worldwide as a surrogate marker for HPV status [15,16]. If the decision-making process strongly relies upon HPV status, the diagnosis of HPV positivity can be further confirmed with the use of in situ hybridization or polymerase chain reaction (PCR) to detect the presence of HPV DNA [17]. Both p16 status and HPV status can be used as standard markers of HPV infection. When the concordance between the two diagnostic tests (detection of protein p16 overexpression by immunohistochemistry and detection of HPV DNA by PCR with genotyping) was analyzed in a population of patients referred for OPSCC, the two tests were 81.7% concordant, with an estimated kappa correlation coefficient of 0.615, indicating a satisfactory concordance according to the Landis and Koch scale [18]. Another biomarker used in clinical practice includes real-time PCR, to measure HPV16 viral load [15]. Less common high-risk HPV subtypes (HPV 18, 31, 33) have also been identified as causative agents of HNSCC and may rarely cause other non-oropharyngeal types of head and neck cancer. 

### 1.3. A Unique Staging System

From a clinical standpoint, a separate staging system was established in 2017 in the 8th edition of the tumor, node, and metastasis (TNM) staging system of the Union for International Cancer Control (UICC)/American Joint Committee on Cancer (AJCC), based on the remarkably better survival outcomes of patients with HPV+ OPSCC compared to those with HPV− OPSCC, reflecting the different biology of the two diseases [19]. The 8th edition adopted the changes in T and N categories proposed by the International Collaboration on Oropharyngeal cancer Network for Staging (ICON-S), with the aim being to “bridge the gap” between the anatomical stage classification and other, more personalized purposes of staging, e.g., prognostication or clinical trial eligibility [20].

## 2. Prognostic Factors and Risk Subgroup Identification

The fact that HPV+ OPSCC has a significantly better prognosis and response to treatment than HPV− OPSCC highlights the emerging necessity to refine patient selection for optimized treatment [1,21]. Identification of the favorable prognostic factors that would lead to patient stratification into risk subgroups could be crucial in determining which patients might be candidates for treatment deintensification, to avoid as much toxicity as possible, without compromising the overall treatment outcome. 

In a recursive partitioning analysis (RPA) of 721 patients with OPSCC by Ang et al. [21], HPV positivity was the strongest independent factor for an improved OS (HPV+ vs. HPV−, hazard ratio (HR) 0.41, 95% confidence interval (CI) 0.29–0.57, *p* < 0.001). A positive history of tobacco smoking and a higher pack-year status were associated with a worse prognosis in patients with HPV+ OPSCC (>10 vs. ≤10 pack-years, HR 1.01, 95% CI 1.00–1.01, *p* < 0.002), as well as a higher nodal stage (N2b-3 vs. N0-N2a, HR 1.68, 95% CI 1.33–2.14, *p* < 0.001). Other covariates with a negative impact on OS were higher age at the time of diagnosis (>50 years vs. ≤50 years, HR 1.35, 95% CI 1.02–1.80, *p* < 0.03), non-white race (non-white vs. white, HR 1.56, 95% CI 1.18–2.05, *p* < 0.002), and a higher tumor stage (T4 vs. T2-3, HR 1.85, 95% CI 1.46–2.34, *p* < 0.001). This RPA stratified patients into three subgroups, where distinct yet homogenous risk of death values were observed; low, intermediate, and high-risk groups with an estimated 3-year OS of 93%, 70.8%, and 46.2%, respectively [21].

In a subset of patients with HPV+ OPSCC presenting with retropharyngeal lymphadenopathy at the time of diagnosis, OS was significantly worse than in patients without this rare location of the nodal disease (5-year OS 57% vs. 81%, HR 2.11, 95% CI 1.04–4.27, *p* = 0.034), without any observed differences in locoregional control (LRC) rates [22].

The presence of a phosphatidylinositol-4,5-bisphosphate 3-kinase catalytic subunit alpha (PIK3CA) mutation worsens the disease-free survival (DFS) in patients with HPV+ OPSCC. Patients with a wild-type-PIK3CA have a statistically significantly higher 3-year DFS than PIK3CA-mutant patients (93.4% vs. 68.8%, 95% CI 85.0–99.9, *p* = 0.004) [23]. 

Absolute lymphocyte count (ALC) might be another valuable prognostic factor in patients with HPV+ OPSCC receiving upfront radiotherapy (RT) with or without concurrent chemotherapy (CHT). In a study of 791 OPSCC patients (67% were HPV+) by Price et al., a higher pretreatment ALC was prognostic for an improved 5-year OS on multivariable analysis (HR 0.64, 95% CI 0.42–0.98, *p* = 0.04). The correlation between pretreatment ALC and OS depended on whether patients had received cisplatin-based CHT or not; the lower the patients´ pretreatment ALC, the higher the potential benefit of cisplatin observed. Patients with a high pretreatment ALC did not benefit from concurrent cisplatin in terms of OS. On the other hand, a low ALC was a poor prognostic factor for patients receiving RT only [24]. 

It has recently been reported that the phenotypic composition of tumor-infiltrating lymphocytes (TILs) may hold additional prognostic value and insight when used as cancer biomarkers. A recent meta-analysis concluded that both CD4+ and CD8+ TILs were associated with a significantly improved survival among all patients with OPSCC, both HPV-negative and HPV-positive, and this topic warrants further focused investigation [25]. 

## 3. Current Treatment Options in Non-Metastatic HPV+ OPSCC by Stage

Currently, the standard treatment recommendations in patients with OPSCC do not differ by HPV status; except in the context of clinical trials. Although testing patients for HPV positivity provides valuable information regarding the patient´s prognosis, there are insufficient data for treatment deintensification based on HPV status only [26]. 

The treatment of HPV+ OPSCC is strictly individual with respect to the stage of the disease and patient´s comorbidities and choice. Curative treatment may include radical surgery, RT, or concurrent chemoradiotherapy (CRT), either as single modalities or in combination. Based on the RPA by Ang et al., low-risk group patients (non-smokers with early-stage disease) are highly curable individuals using a single treatment modality only [21]. Given the fact that long-term survival rates in any locoregionally advanced HPV+ OPSCC may reach up to 80% or higher, the decision-making process should prioritize oropharyngeal function preservation, considering its pivotal role in speech and swallowing, and should emphasize maximum avoidance of late treatment-related adverse events [27,28]. 

### 3.1. Early-Stage (T1-2), Node Negative Disease

In patients with early-stage, node negative disease, either minimally invasive surgery or upfront RT may be offered to patients with expected similar treatment outcomes in institutions with good clinical practice and expertise; the approach would be similar to that of a patient with T1-2N0 HPV− OPSCC [29]. However, in the majority of cases of newly diagnosed HPV+ OPSCC, a small, early-stage primary tumor with gross cervical nodal involvement is present at the time of diagnosis, and the finding of completely negative cervical nodes is unusual [20]. To achieve a complete evaluation of the primary and any nodal disease, computed tomography (CT) or magnetic resonance imaging (MRI) of the neck should image the anatomy, from the skull base to the thoracic inlet. Patients who are scheduled for a definitive RT approach may benefit from the higher sensitivity of fluorodeoxyglucose (FDG) positron emission tomography (PET) CT, for identifying the lymph nodes involved [29]. A recently published study analyzed the role of fluorescence imaging using indocyanine green as a noteworthy alternative technique for sentinel lymph node (SLN) and additional occult regional metastases detection in early-stage node negative (cN0) OPSCC. Utilizing color mode imaging clearly discriminated SLNs from the surrounding structures and the rate of identification of SLNs and false negative rate were 100% and 0%, respectively [30].

#### 3.1.1. Surgery

Surgical treatment as a single modality should be considered when negative margins can be safely obtained (e.g., well-lateralized tonsillar cancers without soft palate involvement), based upon a critical pre-operative assessment of the individual radiological characteristics of the tumor by the treating surgeon. Patients with cancers of the base of the tongue are less appropriate candidates for surgical treatment, due to the very common central location of the lesion and a higher risk of functional impairment of swallowing [29]. Minimally-invasive transoral surgical techniques are preferred rather than open (transmandibular, transcervical) surgical methods. A recently published phase II randomized trial by Ferris et al. reported excellent survival outcomes in patients with HPV+ OPSCC with negative neck lymphatic nodes (LNs) using transoral robotic surgery (TORS) as a single modality. In 38 low-risk patients, the 2-year progression-free survival (PFS) was 96.9% [31]. Transoral robotic surgery and another minimally-invasive surgical technique, transoral laser microsurgery (TLM), are equally effective in early-stage HPV+ OPSCC [32]. In 153 patients treated with primary TLM for OPSCC (66% were HPV+), the 3-year OS, disease-specific survival (DSS), and DFS were 84.5%, 91.7%, and 78.2%, respectively. Patients treated with upfront resection should also receive elective neck dissection (ND), as the probability of occult disease in the regional LNs can be very high, depending on the anatomical site of the primary tumor [33]. Some patients treated with initial surgery may require adjuvant RT or CRT, due to positive margins or extracapsular nodal extension (ENE) or in the case of pathological nodal upstaging. This raises the concern that a patient receiving surgery in the case of an early-stage HPV+ OPSCC may in fact undergo treatment intensification, leading to a risk of permanent treatment-related morbidity. This concern may in fact represent a major drawback of primary surgery, as we are not able to reliably select patients with certainty of R0 resection.

#### 3.1.2. Radiation Therapy

Radiation therapy of the primary tumor is the treatment of choice for patients who wish to avoid potential surgical complications or cannot undergo surgery due to comorbidities or a poor performance status. Elective RT of the neck, eradicating microscopic nodal disease, can be offered bilaterally, or in some cases, unilaterally, without a potential deterioration of the treatment outcome. Ipsilateral neck RT might be preferable in patients with a well-lateralized primary tonsillar HPV+ OPSCC that does not invade >1 cm beyond the mucosa of the soft palate or base of the tongue and that does not extend to the posterior pharyngeal wall [34,35]. Treatment with modern intensity-modulated radiation therapy (IMRT) techniques is considered standard and is well tolerated, with excellent treatment and functional outcomes [31,34]. 

### 3.2. Early-Stage (T1-2), Single Node (≤3 cm) Positive Disease

Several treatment options can be offered to patients with an early-stage (T1-2) primary tumor with a single positive node ≤3 cm. These options include RT only, definitive CRT, or surgery with or without adjuvant RT [29]. The definitive choice of treatment modality should be based upon the individual radiological characteristics of the tumor; its exact anatomic location; local expertise in both minimally-invasive surgery and RT; and the patient´s comorbidities, performance status, and preference. 

#### 3.2.1. Radiation Therapy as a Single Modality

If the disease can be classified as low-risk [21] (i.e., if the patient is a non-smoker with only a single ipsilateral positive LN with no signs of ENE using high-quality imaging), RT can be safely offered as a single treatment modality [36,37,38]. Evidence from single institution series demonstrated the efficacy of IMRT, when precise delineation of the target volume and meticulous planning techniques were used. Based upon a retrospective analysis of 408 patients with OPSCC (p16 status was available in 165 patients (40.4%) out of which 82 patients (66%) were HPV+) by Gutiontov et al. [37], level V lymphatics can be safely omitted from the clinical target volume (CTV), unless there are positive nodes within this anatomical area. The expected risk of regional nodal recurrence in ipsilateral or contralateral level V is 0%, with the vast majority of regional nodal recurrences located in ipsilateral level II [37]. With the sparing of level V lymphatics, RT-related morbidity can be lowered by reducing the risk of long-term dermatitis, brachial plexopathy, and shoulder dysfunction.

#### 3.2.2. Primary CRT

Upfront CRT, rather than RT as a single treatment method, should be considered in smokers or if the positive LN is suspected of ENE. Other high-risk features include an ulcerated primary tumor, some additional clustered and suspicious LNs depicted on high-quality imaging, and cases of retropharyngeal, level IV or V nodal involvement [29]. In patients with HPV+ OPSCC with an early-stage primary tumor with minimum nodal burden, prospective randomized trials are ongoing, to address the risks and potential benefits of RT only vs. CRT, and the definitive treatment of choice remains unclear. Based on the data published from two observational studies [39,40], the survival outcomes are similar for both approaches. In a retrospective study of 1028 OPSCC patients (59% were HPV+) by Hall et al., a 13% benefit in OS for CRT was observed compared to RT only for the HPV+ cohort, but this difference did not reach statistical significance (HR *=* 0.948, 95% CI 0.642–1.400, *p* = 0.91) [40]. Patients treated with CRT are at a higher risk of feeding tube dependency and cisplatin-related toxicity [40,41]. 

#### 3.2.3. Surgery

In a carefully selected subset of T1-2 HPV+ OPSCC patients with a single involved ipsilateral node ≤3 cm (8th edition AJCC stage I disease) and who can achieve satisfactory functional outcomes postoperatively, TORS with excellent oncologic control can be safely offered [42,43,44,45,46]. In contrast to that, if patients pre-operatively evaluated for surgical treatment are found to have high-risk features (radiographic signs of ENE, positive LNs in group IV, V or retropharyngeal LNs, posterior pharyngeal wall, or palatal or mid-tongue involvement), if negative resection margins cannot be secured, or if the resection imposes a threat to the functional outcomes, e.g., it would result in permanent dysphagia or deterioration of patient’s quality of life (QoL), upfront RT or CRT should be considered [29]. Data from a phase II randomized trial of TORS followed by reduced- or standard-dose adjuvant IMRT in resectable HPV+ OPSCC suggest that primary TORS and reduced postoperative RT resulted in outstanding treatment and functional outcomes in intermediate-risk HPV+ OPSCC [31]. Another randomized phase II trial directly comparing the outcomes of TORS plus ND vs. primary RT (ORATOR) in early-stage HPV+ OPSCC revealed evidence of similar treatment and functional outcomes with a distinct toxicity profile [46]. Patients treated with primary RT demonstrated superior swallowing-related QoL scores 1 year after the treatment. There were more cases of neutropenia, hearing loss, and tinnitus reported in the RT group and more cases of trismus and severe dysphagia in the TORS plus ND group [46]. Despite its strengths, ORATOR was a very small study, which only included 68 patients and was conducted across multiple institutions over nearly a decade. Unanswered questions remain about the surgical quality of the trial, given one death among 34 surgical patients (2.9%), compared with one among 495 (0.2%) in the Eastern Cooperative Oncology Group (ECOG) 3311 trial [31]. Given these limitations, the best way to incorporate the results of the ORATOR trial into clinical practice remains challenging [47]. Furthermore, some patients with HPV+ OPSCC that undergo TORS and present with negative LNs or a single positive LN on pre-operative imaging are upstaged in the definitive histopathologic examination and require postoperative RT or CRT. In a retrospective study of 92 patients with OPSCC (84% were HPV+) treated with TORS and ND, 28% of the patients who had no evidence of clinical nodal disease on preoperative imaging had occult pathologic nodal disease, and 22% of the individuals presenting with no nodal disease or with a single positive node on imaging were pathologically upstaged, due to multiple positive LNs [48]. The treating surgeon should therefore discuss the potential advantages and disadvantages of minimally-invasive surgery with the patient in advance, including the possible indication of adjuvant radiation treatment.

#### 3.2.4. Adjuvant RT with or without CHT

As mentioned above, indication of adjuvant RT depends on the presence of high-risk features in the definitive histopathological examination. The omission of adjuvant RT or reduction of its dose remains a topic of investigation. Patients with HPV+ OPSCC with one involved LN >3 cm, and with lymphovascular or perineural spread and close resection margins (≤3 mm), should be offered adjuvant RT rather than observation [29,49,50]. A retrospective analysis of 26 high-risk HPV+ OPSCC patients published by Su et al. suggested that the omission of concurrent CHT with adjuvant RT may offer a comparable LRC with a favorable toxicity profile, in the setting of HPV+ patients and even in those with traditional high-risk features [50]. With a median follow-up of 37 months, none of the patients who received adjuvant RT alone experienced locoregional recurrence. Unilateral IMRT appears to reduce acute toxicity and achieves oncological outcomes similar to those observed in bilateral IMRT in selected patients with lateralized palatine tonsillar cancer with N0-1 nodal disease classification [49]. 

Active smokers may especially benefit from adjuvant RT following TORS. The DSS observed in 15 intermediate-risk HPV+ OPSCC patients in a study published by Miles et al. was 86.7% [51]. 

In the case of positive resection margins, re-resection should be considered, if feasible; while, if patients are ineligible for re-resection or if ENE and >3 positive LNs are present, adjuvant CRT should be indicated [29]. However, the exact number of positive LNs that lead to an indication of adjuvant CRT is unclear and remains controversial. 

Despite the excellent prognosis in HPV+ OPSCC, positive resection margins and ENE are generally associated with the risk of developing distant metastasis [52]. In a retrospective analysis of 296 patients with OPSCC (39% were HPV+) treated with primary surgery, the presence of ENE (*p* = 0.0025) and margin status (*p* = 0.0287) were independent predictors of survival. For HPV+ patients, surgical approach (*p* = 0.0111) and margin status (*p* = 0.0287) were significant predictors of survival, and yet the addition of adjuvant CHT did not confer a survival benefit in the HPV+ subgroup of the study and its exact role in this clinical setting remains a topic of investigation (52).

### 3.3. Locoregionally Advanced Disease (T3-4, Single Positive LN >3 cm, Multiple Positive LNs)

In general, patients with locoregionally advanced HPV+ OPSCC with large primary tumors; and bulky, multiple, and bilateral nodal disease, require multimodal therapy, and the current treatment approach does not significantly differ from that in locoregionally advanced HPV− OPSCC [29]. Primary concurrent CRT (bimodal therapy) should always be preferred over surgical treatment followed by adjuvant CRT (trimodal therapy), as primary CRT offers both organ function preservation and satisfactory treatment outcomes, even in the case of advanced disease. There are few real benefits of trimodal therapy in patients with HPV+ OPSCC, and its indication should therefore be avoided [29]. Importantly, the indication of upfront surgical therapy has been constantly decreasing in this clinical setting [53].

#### 3.3.1. Concurrent CRT

Based on the published data, any RT in combination with CHT in the treatment of HPV+ OPSCC most commonly uses conventional fractionation at 1.8–2.0 Gy per fraction, to a standard dose of 70 Gy in high-risk areas, 54–63 Gy in intermediate-risk areas, and 44–50 Gy in low-risk areas at 7 weeks [29]. Other fraction sizes (e.g., <1.8 Gy or >2 Gy per fraction), the use of simultaneous integrated boost (SIB) IMRT, or concomitant boost acceleration (1.8 Gy/fraction, large field; 1.5 Gy boost as a second daily fraction during the last 12 treatment days) have a similar treatment efficacy; however, there is no consensus on the optimal approach, and the definitive choice of the RT scheme should follow the institutional preference (protocol). The indication of hyperfractionation to a maximum dose of 81.6 Gy in high-risk areas (1.2 Gy/fraction BID, 7 weeks) is currently limited to T2, N0-1 disease only [29]. Concurrent CHT has a substantial additional treatment effect, and acceleration of RT cannot compensate for the absence of CHT [29,54]. Proton beam therapy may be considered when normal tissue constraints cannot be met by a photon-based therapy [29]. 

#### 3.3.2. Concurrent CRT vs. Surgery + Adjuvant CRT

Even though many patients would be staged as having “only” N1 disease based on the 8th edition of the AJCC staging system, the increasing number of positive or matted LNs imposes a dramatically increased risk of local recurrence and distant failure [55,56]. This feature could lead to a potential intensification of RT by concurrent CHT in HPV+ OPSCC patients. 

Randomized data on the use of primary CRT vs. trimodal therapy specifically in HPV+ OPSCC are very limited. In the published retrospective studies, most patients with locoregionally advanced disease treated with upfront surgery were at risk of developing a severe treatment-related functional morbidity, and yet they required additional adjuvant CRT [36,57]. In a retrospective study of 116 HPV+ OPSCC patients treated with ND and either resection of the primary tumor or intraoperative brachytherapy, the 3-year OS was similar between the two groups (91% vs. 87%, *p* = 0.64) [57]. Interestingly, amongst patients with resection of the primary tumor, there was no difference in the 3-year OS between patients with positive vs. negative surgical margins. In a subgroup of patients with ≥5 positive LNs, the predominant pattern of failure was distant, and the rate of distant failure exceeded 50% over the 2-year follow-up [57]. 

#### 3.3.3. The Choice of a Sensitizing Agent with CRT

Although multiple clinical trials and a recently published update on the meta-analysis of chemotherapy in head and neck cancer (MACH-NC) demonstrated a survival benefit with CRT, the optimal concurrent sensitizing agent has not been clearly defined; however, the strongest data support the use of high-dose cisplatin 100 mg/m^2^ [29]. Subset analyses from various prospective randomized trials indicated that a cumulative dose of 200 mg/m^2^ of cisplatin seems to ensure an adequate survival benefit compared to RT alone, though its effectiveness may depend on other factors, such as the frequency of cisplatin administration and RT fractionation pattern [58]. The potential advantages of concurrent cisplatin appear to be age-related, with a loss of benefit in older adult patients [59]. Two main schedules of cisplatin administration are commonly used clinically; bolus administration 100 mg/m^2^ every 3 weeks, and weekly cisplatin 40 mg/m^2^ [29]. In a prospective randomized trial, 270 patients with locally advanced HNSCC (60% of the evaluated patients had OPSCC, out of which only 13% were HPV+) were randomly assigned to definitive CRT, with either weekly cisplatin 40 mg/m^2^ or high-dose (bolus) cisplatin 100 mg/m^2^ every three weeks. With a median follow-up of 26 months, the observed LRC (3-weekly vs. weekly cisplatin 2-year LRC 53% vs. 47%), median PFS (21 months for each arm), and median OS (26 months vs. 30 months) were similar. Severe (grade ≥3) toxicity, including mucositis and nephrotoxicity, was more frequent in patients who had received bolus cisplatin compared to weekly cisplatin [60]. Only with caution can this study be interpreted as proof of the non-inferiority of weekly vs. three-weekly cisplatin, as the treatment outcomes in both arms were rather inferior to other studies and, in most cases, the obsolete 2-dimensional planning was used.

Patients ineligible for cisplatin can receive weekly carboplatin (area under the curve (AUC) 1.5 to 2) [29], but at least one randomized trial concluded that carboplatin was not as effective as high-dose cisplatin [61]. A doublet of carboplatin plus 5-fluorouracil may represent another option when cisplatin is not feasible [62]. When carboplatin and 5-FU are used, the recommended concurrent RT regimen is standard fractionation plus three cycles of this type of CHT [29]. Carboplatin has also been combined with paclitaxel in locally advanced HNSCC [63]. Cetuximab, a monoclonal antibody that inhibits the epidermal growth factor receptor (EGFR), is a less preferable sensitizing agent for patients with good Karnofsky performance status (KPS) and ineligible for cisplatin. While the addition of cetuximab to RT alone demonstrated a significant advantage in terms of OS and LRC in a randomized trial [64], in a subset analysis, its benefit was restricted to patients <65 years with a KPS 90–100, without severe comorbidities [65]. Many elderly patients and those with significant comorbidities are not eligible for concurrent CRT and should be treated with RT alone [29,64,65].

## 4. Treatment Deintensification

Given the excellent prognosis of HPV+ OPSCC following standard treatment, deintensification of therapy (the use of minimally-invasive surgery, de-escalation of adjuvant or primary RT dose, reduction of CHT dose, or substitution of cisplatin by a less toxic drug) in this clinical setting is an area of intensive investigation and research [1,21]. The basic principle is to achieve superior treatment outcomes, while minimizing the risk of treatment-related morbidity. More prospective randomized trials are necessary before the implementation of these less aggressive treatment methods in standard clinical practice. Patients interested in these experimental deintensification approaches are encouraged to participate in clinical trials, if available [66]. 

### 4.1. Minimally-Invasive Surgery

Current data on the use of minimally-invasive surgical methods specifically in HPV+ OPSCC come from small retrospective studies. These studies confirmed the excellent treatment outcomes and safety of TORS [31,45,48,51,52] and TLM plus elective ND [32,33] in early-stage (T1-2) disease with a single positive LN <3 cm. The already mentioned randomized phase II study ORATOR, which compared TORS with primary RT in OPSCC (88% HPV+), provided evidence of similar survival and functional outcomes, with a distinct toxicity profile for each treatment method [46]. In this study, 68 patients were randomly assigned to either TORS plus elective ND (34 patients, 88% HPV+) or primary full-dose (70 Gy) RT (34 patients, 88% HPV+). With a median follow-up of 27 months for the RT group and 29 months for the surgical group, the mean M.D. Anderson Dysphagia Inventory (MDADI) scores were 86.9 in the RT group vs. 80.1 in the TORS plus ND group (*p* = 0.042), but this difference was not clinically meaningful and the swallowing-related QoL of the patients in both arms was similar. One year after the treatment, 100% of the patients in the RT group and 84% in the TORS group continued to receive total oral diet (*p* = 0.055). The OS and PFS were excellent in both groups, and there was no significant difference between the groups. The 3-year OS and PFS for HPV+ patients in the study were 93.1% and 93% [46]. Importantly, among the patients treated with upfront surgery, up to 71% required adjuvant RT or CRT, so whether ORATOR leads to deintensification of the overall treatment remains a topic of debate. 

### 4.2. Dose Reduction of Adjuvant RT

Patients with HPV+ OPSCC that undergo upfront surgery and present with high-risk features (positive resection margins, ENE, multiple positive LNs) in the definitive histopathological examination are indicated to a standard-dose adjuvant RT (54–66 Gy in the intermediate to high-risk areas with 1.8–2 Gy per fraction) [29]. Although some clinical trials have attempted to clarify the significance of reduction of the dose of adjuvant RT, this concept continues to be a subject of investigation and its role remains unclear [31,51,67,68,69,70,71].

Ferris et al. [31] recently published the results of a phase II randomized trial (E3311) comparing TORS followed by a de-escalated adjuvant dose of IMRT in resectable HPV+ OPSCC. In this study, 360 primarily resected patients with stage III-IVA AJCC 7th edition HPV+ OPSCC were assigned to four arms according to risk. Patients with a low-risk disease (11%) (negative resection margins >3 mm, either node-negative (N0) or with a single positive node <3 cm without ENE, no signs of lymphovascular or perineural invasion, arm A) received no further postoperative RT and were observed. Patients with an intermediate-risk disease (58%) (close margins ≤3 mm, 2–4 positive LNs or a single positive LN >3 cm but ≤6 cm, signs of ENE ≤1 mm, positive lymphovascular or perineural invasion) were randomly assigned to receive either de-escalated RT (50 Gy, arm B) or standard-dose adjuvant RT (60 Gy, arm C) [31]. Patients with high-risk disease (31%) (positive margins, ENE >1 mm, ≥5 positive LNs, positive single node >6 cm, arm D) received adjuvant CRT with a standard-dose (60–66 Gy) RT plus weekly cisplatin 40 mg/m^2^. With a median follow-up of 35 months, the 2-year PFS was 96.9% (90% CI 91.9–100) for arm A, 94.9% (90% CI 91.3–98.6) for arm B (50 Gy), 96% (90% CI 92.8–99.3) for arm C, and 90.7% (90% CI 86.2–95.4) for arm D. In an updated report on the E3311 trial, the observed 3-year PFS and OS were 97% and 100% in arm A and 91% and 93% in arm D [67]. Patients treated with a de-escalated dose of adjuvant RT (arm B) demonstrated excellent 3-year PFS and OS (95% and 99%) compared to the patients treated with a standard-dose RT (arm C), where the 3-year PFS and OS reached 94% and 95%. Lower rates of severe (grade ≥3) toxicity were reported in arm B compared to arm C (15% vs. 25%). Furthermore, the evaluated QoL of the patients in arm B using the Functional Assessment of Cancer Therapy—Head & Neck (FACT H&N) scale showed marked improvements when evaluated 6 months after treatment (63%), compared to the patients in arm C (49%) [67]. It can be concluded that TORS alone in low-risk patients and a de-escalated dose of adjuvant RT in intermediate-risk patients demonstrated a remarkable treatment and functional benefit in patients with HPV+ OPSCC, and the evidence provided by the E3311 trial is of significant importance for further randomized trials. Similarly, another phase II randomized trial of 150 HNSCC patients (63% with OPSCC out of which 82% were HPV+) concluded that deintensification of RT, independent of HPV status in a pre-defined low-risk patient population, was safe and resulted in very low rates of late toxicity [68]. De-escalated adjuvant RT that avoids the resected primary tumor bed in patients with negative surgical margins and targets only the at-risk neck areas after TORS might also be a safe deintensification strategy for selected patients with HPV+ OPSCC [69]. 

A single-arm phase II study (MC1273) attempted to determine if aggressive dose de-escalation of adjuvant RT, from a standard dose (60–66 Gy) to 30–36 Gy, for selected patients with HPV+ OPSCC (80 patients) with negative resection margins and a smoking history of 10 pack-years or less could maintain satisfactory treatment outcomes, while reducing toxicity and preserving swallowing-related QoL [70]. Cohort A (intermediate risk patients) received 30 Gy delivered in 1.5 Gy fractions BID over 2 weeks, along with 15 mg/m^2^ docetaxel weekly. Cohort B (high risk patients) received the same treatment, plus a SIB to nodal levels with ENE to 36 Gy in 1.8 Gy fractions BID. For both cohorts, the 2-year LRC, PFS, and OS were 96.2%, 91.1%, and 98.7%, respectively. The rate of severe toxicity at pre-RT and 1- and 2-years post-RT evaluation were 2.5%, 0%, and 0% [70].

Interestingly, based on the results of the MC1273 trial [70], a separate randomized phase III trial MC1675 [71] failed to demonstrate decreased toxicity when applying de-escalated adjuvant RT compared to a standard-dose RT ≥3 months after the treatment. This trial contained 194 resected patients with HPV+ OPSCC with negative margins and no T4 disease. Patients in the MC1675 trial were randomly assigned to receive either a dose-reduced adjuvant CRT (30 Gy for no ENE or 36 Gy for ENE, 1.5 Gy BID plus docetaxel 15 mg/m^2^ weekly) or a standard-dose adjuvant CRT (60 Gy once daily plus cisplatin 40 mg/m^2^ weekly). With a median follow-up of 25 months, the PFS and OS for dose-reduced versus standard-dose RT were 87% and 95%, and 96% and 97%, respectively. The severe toxicity ≥3 months after RT (primary endpoint) for dose-reduced versus standard-dose RT was 2% vs. 7% [71]. Dose-reduced adjuvant RT improved the overall patient-reported QoL, including swallowing, pain, and xerostomia items one month after RT. With a median follow-up of 25 months, the PFS and OS for dose-reduced vs. standard-dose RT were 87% vs. 95% and 96% vs. 97%, respectively [71]. A longer follow-up time in the presented studies and more prospective randomized trials, including a higher number of patients, are necessary to critically assess the potential benefits of reduced-dose RT in this clinical setting. 

### 4.3. Dose Reduction of Primary RT

In an attempt to preserve organ function, avoid surgical complications and potentially achieve excellent treatment outcomes with decreased toxicity, some studies have focused on reduction of the primary RT dose. A randomized phase II study (ORATOR2) compared upfront TORS plus ND followed by a reduced-dose adjuvant RT (50 Gy in 25 fractions or 60 Gy in 30 fractions if there were positive margins or ENE) with a primary de-escalated RT (60 Gy) with weekly cisplatin 40 mg/m^2^ [72]. Overall, 61 patients with T1-2 N0-2 AJCC 8th edition HPV+ OPSCC were randomized; 31 in the TORS group and 30 in the primary RT group. Patients treated with upfront de-escalated RT demonstrated superior results for 2-year OS and PFS compared to the TORS group (100% vs. 89%, and 100% vs. 84%). The median follow-up in ORATOR2 was only 19 months, and the patient accrual was halted due to the excessive toxic effects observed in the study. Two treatment-related deaths were reported after receiving TORS in the trial: one due to oropharyngeal bleeding, and another where the patient died due to cervical vertebral osteomyelitis (spondylodiscitis). As both Grade 5 events are rather unusual and as the percentage of treatment-related deaths in this study was 3.2%, questions have been raised about the quality of the surgery and RT in the trial [47]. Patients in both treatment arms in ORATOR2 achieved good swallowing outcomes at one year after the treatment. Grade 2–5 adverse events occurred in 67% of the patients in the RT group compared to 71% in the TORS group [72]. Careful patient selection and high-level surgical expertise in high-volume centers are of paramount importance for an uncomplicated procedure of TORS in HPV+ OPSCC.

In a non-randomized phase II trial published by Chera et al., 43 patients with T0-3 N0-2c (AJCC 7th edition) HPV+ OPSCC with minimal smoking history were treated with de-escalated IMRT (60 Gy) with weekly concurrent cisplatin 30 mg/m^2^. The complete response (CR) rate was 86% in this study, evaluated by biopsy of the primary tumor and a ND performed 6–14 weeks after RT [73,74]. Secondary endpoint measures included physician-reported adverse events (Common Toxicity Criteria for Adverse Events, CTCAE) and patient-reported symptoms (patient reported outcomes, PRO-CTCAE). The incidence of CTCAE severe toxicity and PRO-CTCAE severe symptoms in the individually evaluated items was as follows: mucositis 34%/45%, general pain 5%/48%, nausea 18%/52%, vomiting 5%/34%, dysphagia 39%/55%, and xerostomia 2%/75% [73]. The 3-year local control, regional control, cause-specific survival, and OS were 100%, 100%, 100%, and 95%, respectively [74]. The authors concluded that a substantial decrease in the intensity of RT and a reduced weekly low-dose cisplatin produced better preservation of QoL compared to standard therapies, while achieving excellent 3-year tumor control and survival. The reported survival and toxicity outcomes from other published studies are listed in Table 1 [74,75,76,77,78,79,80]. 

### 4.4. Omission of Concurrent CHT

In a randomized phase II NRG-HN002 trial, 306 patients with T1-T2 N1-N2b and T3 N0-N2b HPV+ OPSCC (AJCC 7th edition) with ≤10 pack-years history of smoking received either 60 Gy IMRT over 6 weeks with concurrent weekly cisplatin 40 mg/m^2^ (arm A) or 60 Gy accelerated IMRT over 5 weeks without CHT (arm B) [79]. The 2-year PFS and OS were 90.5% and 96.7% vs. 87.6% and 97.3% in arms A and B, respectively. There were more significant severe acute adverse events in arm A compared to arm B (79.6% vs. 52.4%; *p* < 0.001). Higher rates of 2-year locoregional failure were recorded in arm B compared to arm A (10% vs. 3%). Maximal local control is the reason why many studies use CRT rather than RT alone as the control arm in studies evaluating treatment deintensification in HPV+ OPSCC. 

### 4.5. Dose Reduction of Chemosensitizing Agent with RT

A high proportion of patients with low-risk HPV+ OPSCC may benefit from a reduction of the concurrent cisplatin added to RT. In a single arm, non-randomized phase II trial published by Chera et al., 114 patients with T0-3 N0-N2c (AJCC 7th edition) HPV+ OPSCC and a limited smoking history (<10 pack-years) were treated with primary de-escalated definitive RT (60 Gy, 6 weeks), with or without reduced concomitant cisplatin 30 mg/m^2^ administered weekly (6 applied doses) [80]. Patients with a low-risk disease (T0-2 N0-1) received RT alone, while patients with an intermediate- to high-risk disease received CRT with reduced doses of cisplatin. All patients in the study underwent PET/CT evaluation 12 weeks after the treatment, to assess for ND. With a median follow-up of 31.8 months, the 2-year LRC, PFS, and OS were 95%, 86%, and 95%, respectively. The reported pre- and 2-year post-treatment European Organization for Research and Treatment of Cancer (EORTC) QoL scores were as follows: global, 79/84 (lower worse); swallowing, 8/9 (higher worse); and dry mouth, 14/45 (higher worse). The reported pre- and 2-year post-treatment PRO-CTCAE QoL scores (0 to 4 scale, higher worse) were as follows: swallowing 0.5/0.7; and dry mouth 0.4/1.3. No severe late toxicity was observed in the trial [80]. These encouraging study results demonstrate that de-intensified primary RT with reduced concurrent CHT may result in good oncological outcomes, with a high QoL reported by the patients, and that it is worthy of further clinical investigation.

### 4.6. Substitution of Concurrent Cisplatin with Cetuximab

Cetuximab potentially offers a less toxic alternative to cisplatin. Two large prospective randomized phase III trials were conducted with a similar aim; to assess the current position of cetuximab within the definitive oncological treatment of HPV+ OPSCC patients. The first study, published by Mehanna et al. in 2019 (DE-ESCALATE), evaluated 334 eligible low-risk patients with no or limited smoking history (<10 pack-years) [81]. Eligible patients were randomly assigned to receive either intravenous cisplatin (100 mg/m^2^ on days 1, 22, and 43, 166 patients) or intravenous cetuximab (400 mg/m^2^ loading dose followed by seven weekly administrations 250 mg/m^2^, 168 patients) in combination with standard-fractionated RT (70 Gy in 35 fractions). The 2-year observed severe toxicity (the primary end-point of the study) did not differ significantly between the two groups. However, there was a significant difference in 2-year OS in the cetuximab vs. cisplatin group (89.4% vs. 97.5%, HR 5.0 [95% CI 1.7–14.7], *p* = 0.001) and 2-year recurrence rate (16.1% vs. 6.0%, HR 3.4 [95% CI 1.6–7.2]; *p* = 0.0007) [81].

The second study, the non-inferiority trial RTOG 1016, recruited 849 patients with HPV+ OPSCC (AJCC 7th edition) T1-T2 N2a-N3 M0 or T3-T4 N0-N3 M0 stage [82]. Patients were randomly assigned to receive either intravenous cetuximab (loading dose 400 mg/m^2^ followed by cetuximab 250 mg/m^2^ weekly, 7 doses, 425 patients), or cisplatin 100 mg/m^2^ on days 1 and 22 of RT (total cumulative dose 200 mg/m^2^, 424 patients). All patients received accelerated IMRT (70 Gy in 35 fractions over 6 weeks, 6 fractions per week, 2 two fractions given on one day, 6 h apart). The observed 5-year OS (the primary end-point of the study) was 77.9% in the cetuximab group vs. 84.6% in the cisplatin group. The 5-year PFS was significantly lower in the cetuximab group compared to the cisplatin group (67.3% vs. 78.4%, HR 1.72, 95% CI 1.29–2.29; *p* = 0.0002). Furthermore, the 5-year locoregional failure was significantly higher in the cetuximab group compared with the cisplatin group (17.3% vs. 9.9%). The observed acute and late moderate to severe toxicity rates were similar in both groups [82].

These trials confirmed a significant OS and LRC benefit in favor of cisplatin. Interestingly, when the data for low-risk patients in DE-ESCALATE were analyzed, there was still a significant absolute difference of 5.2% in 2-year OS in favor of cisplatin [81,83]. It can be concluded that a cisplatin dose of 100 mg/m^2^ every 3 weeks with 70 Gy of RT is the regimen supported by the most robust evidence and remains the standard of care [29,81,82,83,84,85,86]. High-risk patients who are heavy smokers (>10 pack-years) or those with T4 and N3 disease demonstrate significantly poorer outcomes than other patients with HPV+ OPSCC and should not be considered candidates for de-escalation strategies [82,83,84].

### 4.7. Induction CHT Followed by Dose-Reduced RT

Partial or complete treatment response to induction CHT (iCHT) in HNSCC is a good prognostic factor [87]. For iCHT responders, CRT is usually offered with the aim of achieving organ preservation and high QoL of the patients; and, besides, surgery and CRT achieve similar survival probabilities in those who respond to iCHT [87]. Certain patient subgroups that do not respond to iCHT demonstrate inherent chemo- and radioresistance. These patients should be referred for surgical treatment, if feasible, with little or no expected benefit from the following RT. 

As the concept of local and regional control improves due to the implementation of novel technologies in bimodal treatment (TORS, IMRT, image-guided RT—IGRT), distant metastasis is increasingly emerging as one of the major causes of treatment failure. Therefore, iCHT aimed at improving distant control may increase overall treatment success. The present evidence contradicts the hypothesis that iCHT could improve OS by lowering the rate of distant failure [88]. Despite decades of investigation, the exact benefits of iCHT and subsequent RT remain unclear, except for in organ preservation in patients that would otherwise require total laryngectomy. Furthermore, given the toxicities of iCHT, it is uncertain to what extent this treatment concept can be referred to as “deintensification”.

Some published data coming from single-arm non-randomized phase II trials suggested that iCHT may allow for subsequent de-escalated definitive RT. In ECOG 1308, 80 patients with stage III-IVA AJCC 7th edition HPV+ OPSCC received three cycles of iCHT (cisplatin, paclitaxel, cetuximab) [89]. Complete responders proceeded to de-escalated RT (54 Gy in 27 fractions), and those with less than CR at the primary site or LNs proceeded to a standard-dose (69.3 Gy) RT to those regions. In both groups, RT was administered concurrently with weekly cetuximab. Complete response was reported in 70% of the patients. Fifty-one patients (64%) continued to IMRT 54 Gy plus cetuximab (arm A); and in 27 patients (34%), standard-dose RT plus cetuximab was administered (arm B). The 2-year PFS and OS were 80% vs. 96%, and 94% vs. 96%, in arm A vs. B [89]. Patients with T4 classification disease and those with >10 pack-years were much more likely to experience recurrence. Reduction of RT dose led to a significantly improved swallowing and nutritional status. 

The results from ECOG 1308 resemble the conclusions drawn from an earlier trial, ECOG 2399. However, ECOG 2399 contained various HNSCC patients (105), and the proportion of OPSCC was only 66% [90]. Correlation of treatment response and outcomes, with regard to HPV status, was presented separately by Fakhry et al. [91]. In ECOG 2399, patients received two cycles of iCHT (carboplatin plus paclitaxel) and proceeded to a standard-dose (70 Gy) RT with concurrent weekly paclitaxel 30 mg/m^2^. Patients with HPV+ compared to HPV− OPSCC demonstrated superior results in response rates following iCHT (82% vs. 55%) and definitive CRT (84% vs. 57%), as well as 2-year OS (95% vs. 62%) [91].

Another single-arm phase II study published in 2017 by Chen et al. enrolled 44 patients with AJCC 7th edition stage III-IV HPV+ OPSCC [92]. The patients received two cycles of iCHT (paclitaxel 175 mg/m^2^ and carboplatin AUC 6), followed by IMRT 54 Gy in 27 fractions in partial or complete responders (55%), and 60 Gy in 30 fractions for those that had less than partial or no response to iCHT (45%). Radiation treatment was administered daily with IGRT and concurrently with paclitaxel 30 mg/m^2^ weekly. With a median follow-up of 30 months, the 2-year PFS was 92% for all patients enrolled in the study and was not calculated separately for the two patient subgroups receiving distinct doses of RT. Thirty-nine percent of the patients had grade 3 adverse events [92]. The authors concluded that the reduction of RT doses by 15–20% following iCHT was associated with a high PFS and an improved toxicity profile, in comparison with the historical regimens using standard RT doses. 

In a single-arm phase II trial (OPTIMA), 62 patients with stage III-IV AJCC 7th edition HPV+ OPSCC received three cycles of iCHT (carboplatin AUC 6 and nab-paclitaxel 100 mg/m2) followed by dose- and volume-reduced 1.5 Gy BID RT (every other week) or CRT with paclitaxel, fluorouracil and hydroxyurea [93]. Patients were classified either as low-risk (≤T3, ≤N2B, ≤10 pack-year smoking status) or high-risk (T4 or ≥N2C or >10 pack-year smoking status). Low-risk patients with ≥50% response received 50 Gy RT (arm A), and low-risk patients with 30%–50% response or high-risk patients with ≥50% response received 45 Gy CRT (arm B). Patients with a less than partial or no response received 75 Gy CRT (arm C). Interestingly, within OPTIMA, RT was volume-limited to the first echelon of uninvolved LNs. With a median follow-up of 29 months, the reported 2-year PFS and OS were 95% and 100% for low-risk patients and 94% and 97% for high-risk patients. Severe mucositis occurred in 30%, 63%, and 91% in arms A, B, and C [93]. 

The exact role of iCHT in HPV+ OPSCC needs to be evaluated in prospective randomized trials, before making any definitive conclusions about its benefits. Of note, the initial response to iCHT remains a valuable prognostic biomarker, and the results drawn from single-arm phase II studies incorporating risk and response adaptive locoregional treatment are worthy of further clinical attention, with regard to organ function preservation and acceptable toxicity.

### 4.8. Current Practices in Deintensification 

The extent of primary RT dose-reduction (<66 Gy) in the United States was the key-point of a retrospective database review of 617 HPV+ OPSCC patients [94]. De-escalated RT was delivered in 16.9% HPV+ OPSCC patients, predominantly the elderly. Among HPV+ OPSCC patients, the reported 3- and 5-year OS rates were 83% and 80% in the de-escalated cohort vs. 83% and 78% in the standard-dose cohort (*p* = 0.83). The utilization of de-escalated primary RT in the United States is approximately 15–20% and does not seem to impact survival in HPV+ OPSCC patients.

To investigate the current patterns of deintensification of the treatment of HPV+ OPSCC in South Korea, 42 head and neck oncology experts from the Korean Society for Head and Neck Oncology were surveyed using a questionnaire [95]. Approximately 20% of the respondents prescribed radiation dose reduction in the tonsillar fossa and high-risk cervical LNs in primary CRT of HPV+ OPSCC; however, no tendency to reduce the radiation field was observed. For stage T2N1M0 OPSCC, postoperative RT was reduced to ≤50 Gy for the ipsilateral tonsil and involved neck by 21% of the treating specialists. In complete responders following iCHT, reduction of definitive CRT was performed by 19% of the surveyed oncologists [95]. Consensus guidelines might be established in the near future after a detailed analysis of the results of ongoing prospective trials.

#### Risk Stratification for Treatment Deintensification following TORS According to the Resection Margin Status

In recent randomized trials evaluating the exact role of TORS in the treatment of HPV+ OPSCC patients, the resection margin status (negative, close, positive) has been identified as a major risk factor for treatment deintensification and serves as a pivotal tool for patient stratification into risk groups receiving control or test treatments. However, the exact definition of negative, close, and positive margins differed substantially across the individual studies [31,46,72,96]. The E3311 trial [31] defined negative resection margins as >3 mm, close margins as <3 mm, and positive margins as <1 mm, which, among other risk factors, stratified the patients into low-, intermediate-, and high-risk groups, where observation, adjuvant RT 50 or 60 Gy (random allocation), or adjuvant CRT with 66 Gy was indicated [31]. Interestingly, the ORATOR study defined close resection margins as <2 mm, which classified patients into the intermediate-risk group with 60 Gy of adjuvant RT [46]. The ORATOR2 study [72] defined close resection margins as <3 mm, similarly to the E331 trial [28]. Furthermore, the currently ongoing PATHOS trial [96] defines a positive margin as <1 mm, leading to a higher dose of administered postoperative RT (at least 60 Gy with or without CHT). A close resection margin is defined as 1–5 mm, where patients are randomized to receive either 50 or 60 Gy of adjuvant RT. This substantial variability of the definition of close surgical margins carries the risk of under- or overtreatment of some patients and remains a topic of controversy.

## 5. Ongoing Clinical Trials

Deintensification of the treatment in HPV+ OPSCC is an emerging topic that has been gaining attention worldwide, not only due to the increasing incidence of the disease, resembling an epidemic in urban areas in many developed countries, but also due to its excellent prognosis, with respect to its very distinct and unique biological properties. Many clinical trials have attempted to clarify the potential benefits of either RT dose de-escalation or concurrent CHT dose reduction, with the addition of novel treatment methods, e.g., concurrent immunotherapy, being evaluated.

A currently ongoing phase III trial (NRG-HN005; NCT03952585) randomly has assigned non-smoking HPV+ OPSCC patients to either standard-dose (70 Gy) RT with high-dose cisplatin or dose-reduced RT (60 Gy) administered concurrently with bolus cisplatin or the immune checkpoint inhibitor, anti-programmed cell death protein (PD-1), nivolumab. The primary end-points of the study are PFS and global QoL, evaluated with the use of the MDADI scale [97]. The addition of immunotherapy and its potential benefits in intermediate-risk locoregionally, advanced HPV+ OPSCC is the key-point of another, randomized phase II trial (NCT03410615) [98]. In this trial, treatment outcomes of bolus cisplatin (100 mg/m^2^ days 1, 22, 43) plus standard-dose (70 Gy) RT are being compared to durvalumab, a monoclonal checkpoint inhibitor that blocks the interaction of programmed cell death ligand 1 (PD-L1) plus 70 Gy of RT, followed by adjuvant durvalumab or the combination of tremelimumab (another monoclonal checkpoint inhibitor that blocks downregulation of T-cell activation by binding to the cytotoxic T-lymphocyte antigen-4 receptor, CTLA-4) and durvalumab. The primary end-point of the study is 3-year event-free survival.

Another phase II trial (NCT03799445) is investigating the outcomes of de-escalated (50–66 Gy) RT in the treatment of low- to intermediate-volume locoregionally advanced HPV+ OPSCC, administered in combination with another anti-CTLA4 inhibitor, ipilimumab, and anti-PD-1 checkpoint inhibitor nivolumab. The primary end-points of this study are PFS, CR rate, and the occurrence of dose-limiting toxicity [99]. 

A randomized phase II (SIRS 2.0; NCT05419089) trial has stratified early and intermediate stage HPV+ OPSCC patients with PCR detectable plasmatic cell-free HPV DNA (cfHPV DNA) following TORS into risk groups based on final histopathological features of the disease, to determine the appropriate postoperative risk-adapted treatment intensity [100]. Patients with low-risk pathologic disease and undetectable postoperative cfHPV DNA are receiving no adjuvant therapy and are observed only. Patients with high-risk pathologic disease and undetectable postoperative cfHPV DNA are receiving dose-reduced adjuvant CRT. The primary end-point of this study is 2-year local or regional recurrence rate [100]. 

## 6. Future Perspectives

The results of many trials published to date investigating the role of deintensification have been encouraging; however, well-designed phase III trials are of utmost importance to potentially establish a new standard of care for patients with HPV+ OPSCC. The future will likely hold various therapeutic options offering similar outcomes following de-escalation, instead of a single recommended treatment strategy. 

### 6.1. Novel Biomarkers

Novel biomarkers such as plasma cell-free HPV DNA and genomic data (PIK3CA mutation) are likely to gain importance for risk-stratification in the future generation of trials [101,102,103,104]. 

#### 6.1.1. Cell-Free HPV DNA

Some evidence suggests that the plasmatic levels of cell-free (i.e., circulating tumor) HPV-DNA correlate with the size of the primary tumor and its stage before any therapy is initiated in HPV+ OPSCC patients [101]. Increased plasmatic levels of cfHPV DNA have been reported in patients who develop a clinically obvious relapse after treatment; while in patients in clinical remission, the levels of cfHPV DNA remain undetectable [96]. In a prospective study published by Chera et al., blood samples, testing the presence of cfHPV DNA in 115 patients with HPV+ OPSCC treated with primary CRT, were obtained every 6–9 months. The negative predictive value of cfHPV DNA in the detection of disease recurrence was 100%. The positive predictive value for recurrence derived from two consecutive positive tests was 94% [102]. Interestingly, the phenomenon of rapid plasmatic clearance of cfHPV DNA in HPV+ OPSCC patients treated with definitive CRT correlates with the clinical control of the disease [103]. In a prospective biomarker trial conducted in 103 patients with HPV+ OPSCC patients, blood specimens were collected at baseline, weekly during CRT, and during follow-up visits. Baseline plasma cfHPV DNA levels were highly specific (97%) and highly sensitive (89%) in detecting newly diagnosed HPV+ OPSCC. A favorable clearance profile was defined as having a high baseline copy number (>200 copies/mL) and >95% clearance of cfHPV DNA within the fourth week of CRT. In 28% of the patients, a favorable cfHPV DNA clearance profile was identified, and none of these patients had persistent or recurrent disease after CRT [103]. A rapid clearance profile of cfHPV-DNA may be useful for risk stratification and patient selection for next-generation treatment deintensification.

#### 6.1.2. PIK3CA Mutation 

The most frequently mutated gene in HPV+ OPSCC is the PIK3CA gene [23]. To clarify its prognostic significance, 77 patients with HPV+ OPSCC were enrolled in two phase II clinical trials of deintensified CRT (60 Gy IMRT plus weekly cisplatin). Patients with wild-type-PIK3CA had statistically significantly higher 3-year DFS than PIK3CA-mutant patients (93.4% vs. 68.8%, *p* = 0.004). In a multivariate analysis, PIK3CA mutation was the only variable significantly associated with the risk of disease recurrence [23]. The PIK3CA mutation is associated with worse DFS in HPV+ OPSCC patients, and its presence could serve as an exclusion criterion from future clinical trials aiming at treatment deintensification. 

### 6.2. Increased Radiosensitivity

Human papillomavirus causes carcinogenesis by expressing proteins E6 and E7. However, the expression of E6 and E7 itself is inadequate to cause malignant transformation of the epithelial cells. The DNA methyltransferase 1 (DNMT1) encodes an enzyme that transfers methyl groups to cytosine nucleotides of the genomic DNA [104]. Zhang et al. first demonstrated that DNMT1 downregulated the expression of serine/threonine-protein kinase-1 (SMG1), an enzyme that is encoded by the SMG1 gene in humans, and concluded that this was the underlying mechanism leading to an increased sensitivity of HPV+ OPSCC cells to RT [104]. The study determined that HPV E6 could downregulate the expression of SMG1, a potent tumor suppressor, by upregulating the expression of DNMT1, in order to decrease DNA stability and facilitate the effects of RT. By regulating the expression of SMG1, which seems to be an effective switch in RT sensitivity regulation, HPV+ OPSCC patients could benefit more from RT, which could, in turn, result in their improved survival. 

Overexpression of EGFR is another mechanism affecting the radiosensitivity of HPV+ OPSCC cell lines [105]. In response to RT, EGFR activation was followed by inactivation of the DNA double-strand break repair mechanisms, which resulted in an increased sensitivity to RT. At the same time, EGFR was found to downregulate E6 expression and induced p53 activity in response to RT [105]. However, the addition of an EGFR inhibitor, Cetuximab, to primary RT demonstrated inferior survival results in HPV+ OPSCC patients, in comparison to patients treated with the standard-of-care cisplatin combined with definitive RT [81,82,83,84]. Definition of the exact pathways by which EGFR downregulates the expression of E6 protein could yet provide a new insight into the molecular pathogenesis of HPV+ OPSCC.

### 6.3. Proton-Based RT

Targeting the DNA double-strand break repair mechanisms further enhances the radiosensitivity of HPV+, as well as HPV− HNSCC to photons and protons. The response to radiosensitization by targeting the major protein kinases involved in the DNA double-strand break repair signalization pathway, namely ataxia telangiectasia-mutated (ATM), ataxia telangiectasia, and Rad3-related (ATR), and the catalytic subunit of DNA-dependent protein kinase (DNA-Pkcs), with the use of both photons and protons. was analyzed by Vitti et al. [106]. The inhibition of ATM, ATR, and particularly DNA-Pkcs, caused a significant reduction of HPV-positive and -negative HNSCC cell in vitro model survival when both photons and protons were used. Interestingly, HPV+ OPSCC cell lines showed strongly enhanced radiosensitivity after photon, but not after carbon ion, irradiation [107].

An increasing use of proton beam RT for the treatment of HPV+ OPSCC is being observed worldwide. To date, there is no evidence randomly comparing the effect of photon and proton RT in this clinical setting. Intensity-modulated proton therapy (IMPT) itself is a form of RT de-escalation, by reducing the dose to organs at risk and normal healthy tissues, without compromising the dose delivered to the tumor and regional LNs [108]. Retrospective studies comparing photon-based IMRT to IMPT for OPC suggest similar results for treatment outcomes and lower rates of toxicity (pain, xerostomia, dysphagia, or feeding tube dependence), all of which may have a positive impact on patient´s QoL [109,110,111,112,113,114,115,116,117,118] (Table 2). A model-based selection of patients with OPSCC for proton therapy is also clinically feasible. In a study published by Tambas et al., approximately one third of all HNSCC patients qualified for proton therapy, and these patients had the highest benefit from IMPT, in terms of toxicity prevention [119]. 

Two currently ongoing phase III trials, the “Intensity-Modulated Proton Beam Therapy or Intensity-Modulated Photon Therapy in Treating Patients With Stage III-IVB Oropharyngeal Cancer” trial [120] and the “Toxicity Reduction Using Proton Beam Therapy for Oropharyngeal Cancer (TORPEDO)” trial [120] are aiming to provide level I evidence on the efficacy of IMPT in the treatment of HPV+ OPSCC. The ongoing clinical trials are listed in Table 3.

## 7. Conclusions

While the field of head and neck oncology remains an active focus of research, it seems that those studies directed towards HPV+ OPSCC are becoming the focus of both basic and clinical scientists. This is a consequence of the increasing incidence of these cancers, as well as the accumulated evidence pointing to a better prognosis of this subgroup of patients when compared to their HPV− counterparts. It is, therefore, not surprising that research efforts are mostly directed towards the “optimization” of treatment approaches, which many clinicians consider synonymous with treatment deintensification in HPV+ OPSCC. The early results, obtained predominantly from phase II studies, are promising and suggest that, with careful patient selection, de-escalated primary or adjuvant RT schemes may result in non-inferior treatment results. Importantly, cetuximab-based de-escalation strategies failed to demonstrate a therapeutic benefit in three phase III studies, highlighting the pivotal role of cisplatin in the survival of HPV+ OPSCC patients. New horizons and standards of care for HPV related OPSCC patients will depend on the precise interpretation of each clinical trial aiming at treatment deintensification. Dozens of clinical trials investigating the benefits of many types of deintensified treatment protocols are currently ongoing, of which many are using variable inclusion criteria that will make the potential results difficult to extrapolate to a broad spectrum of HPV+ OPSCC patients. While some trials enroll patients according to their stage and limit the selection to early-stage tumors only, other trials guide the selection by smoking history and also include individuals with locoregionally advanced disease. The cost-effectiveness of novel biomarkers and treatment strategies, e.g., routine cell-free HPV DNA determination or proton therapy might prohibit their routine implementation into standard clinical practice. Finally, not only the functional evaluation of adverse events by the treating physician, but also patient-reported quality of life measures will provide exciting insights into the treatment-related morbidity and functioning of the survivors who have undergone treatment. 

## Figures and Tables

**Table 1 cancers-14-05385-t001:** Primary de-escalated CRT in the treatment of HPV+ OPSCC.

	No. of Patients	RT Dose (Gy)	CHT Dose (mg/m^2^)	p/y	Median FU	Treatment Response (%)	AE	QoL Evaluation
Chera [74] 2018	44	60 HR 54 IR	cisplatin, 30	≤10	36 mo	pCR 86 3-y LC 100 3-y RC 100 3-y CSS 100 3-y OS 95	acute 39 late 0	EORTC QLQ PRO-CTCAE
Woody [75] 2016	50	70–74.7 primary 54 INA	cisplatin, 100 cisplatin + 5-FU cetuximab	4–90	54 mo	5-y LRC 96 5-y DFS 81 5-y OS 86	N/A	N/A
Sher [76] 2021	51	64 INA 40 ENA	cisplatin, 100 cisplatin, 40 carboplatin + paclitaxel cetuximab	≤10 >10	24.7 mo	2-y LRR 12 2-y OS 91 2-y PFS 81	acute 33 late 1.9	EORTC QLQ MDADI EQ-5D
Deschuymer [77] 2020	82	70 HR 40 ENA (arm A) 50 ENA (arm B)	N/A 65% in arm A 71.5% in arm B	N/A	7.6 y	arm A: 5-y RR 14 5-y LRR 24.7 5-y OS 56.5 arm B: 5-y RR 7.5 5-y LRR 17 5-y OS 49.6	N/A	N/A
Tsai [78] 2022	276	70 HR 50 IR 30 LR	cisplatin, 100 carboplatin + paclitaxel carboplatin + 5-FU cetuximab	≤10	26 mo	2-y LRC 97 2-y PFS 88 2-y OS 95.1	acute 6.2	GTQ EORTC QLQ
Yom [79] 2021	158 (arm A) 150 (arm B)	60 HR 54 IR 48 LR	cisplatin, 40 (arm A) no CHT (arm B)	≤10	2.6 y	arm A: 2-y PFS 90.5 2-y OS 96.7 arm B: 2-y PFS 87.6 2-y OS 97.3	arm A: acute 79.6 late 21.3 arm B: acute 52.4 late 18.1	MDADI
Chera [80] 2019	114	60 HR 54 IR	cisplatin, 30 cetuximab, 250 carboplatin AUC 1.5 + paclitaxel 45	≤10	31.8 mo	CR 93 primary 80 nodal 2-y LRC 95 2-y PFS 86 2-y OS 95	acute 34 late 0	EORTC QLQ PRO-CTCAE

Abbreviations: no.—number; RT—radiation therapy; Gy—Gray; HR—high-risk; IR—intermediate risk; LR—low risk; INA—involved nodal areas; ENA—elective nodal areas; CHT—chemotherapy; 5-FU—5-fluorouracil; AUC—area under the curve; N/A—not available; FU—follow-up; mo—months; y—years; pCR—pathological complete response; CR—complete response; LC—local control; RC—regional control; CSS—cause-specific survival; OS—overall survival; LRC—locoregional control; DFS—disease-free survival; RR—response rate; LRR—locoregional recurrence rate; PFS—progression-free survival; QoL—quality of life; EORTC QLQ—European Organization for Research and Treatment of Cancer Quality of Life Questionnaire; PRO-CTCAE—Patient Reported Outcomes—Common Terminology Criteria for Adverse Effects; GTQ—Gothenburg Trismus Questionnaire; MDADI—M.D. Anderson Dysphagia Inventory, EQ-5D—EuroQol 5-Dimension scale questionnaire, p/y—Smoking history (pack-years), AE—Severe (≥Grade 3) adverse events (%).

**Table 2 cancers-14-05385-t002:** Proton-based RT in the treatment of HPV+ OPSCC.

	No. of Patients	RT Dose (GyE)	Applied CHT (%)	Median FU	Treatment Response (%)	Severe (≥Grade 3) Adverse Events (%)
Slater [109] 2005	29	75.9/45 fr	N/A	28 mo	2-y LRC 93 2-y DFS 81 5-y LRC 84 5-y DFS 65	late 11 2-y late 16
Frank [110] 2014	15	66–70/33 fr	80	28 mo	CR 93.9	acute 40 late 7
Gunn [111] 2016	50	60–70	64	29 mo	2-y PFS 88.6 2-y OS 94.5	acute 58 late 12
Bahig [112] 2019	103	N/A	70	3.3 y	3-y LRC 93 3-y OS 96 5-y LRC 90 5-y OS 80	acute 46
Aljabab [113] 2020	46	60–74.4	64	19.2 mo	CR 100 primary CR 92 nodal 1-y LRC 100 1-y OS 95.7	acute 76 late 0
Blanchard [114] 2016	50	66–70	33	29 mo	3-y LRC 91 3-y PFS 86.4 3-y OS 94.3	acute 40.8 late 42
Zhang [115] 2017	50	70/33 fr	64	34.6 mo	N/A	N/A
Manzar [116] 2020	46	60–70	78.3	12 mo	N/A	acute 23.3
Cao [117] 2021	103	66–70	N/A	36.2 mo	N/A	2-y late 6 3-y late 6
Sio [118] 2016	35	67–70	N/A	7.7 mo	N/A	N/A

Abbreviations: no.—number; GyE—Gray Equivalent; RT—radiation therapy; CHT—chemotherapy; FU—follow-up; mo—months; y—years; fr—fractions; N/A—not available; CR—complete response; OS—overall survival; LRC—locoregional control; DFS—disease-free survival; PFS—progression-free survival.

**Table 3 cancers-14-05385-t003:** Ongoing clinical trials in HPV+ OPSCC.

Trial Registration Number	Brief Description	Estimated No. of Patients	Primary End-Point	Secondary End-Points
NCT03952585 [97]	randomized II/III de-intensified RT with CHT (cisplatin) or immunotherapy (nivolumab)	711	PFS QoL	LRF DF OS incidence of AE
NCT03410615 [98]	randomized II cisplatin + RT vs. durvalumab + RT followed by durvalumab vs. durvalumab + RT followed by tremelimumab + durvalumab	180	3-y EFS	FACT-HN score LRF DMFS OS
NCT03799445 [99]	non-randomized II ipilimumab, nivolumab and RT	180	DLT CRR PFS	incidence of acute and chronic AEs
NCT05419089 [100]	non-randomized non-inferiority The Sinai Robotic Surgery Trial (SIRS 2.0 Trial)	199	LRR	PFS DFS OS MDADI score
NCT01893307 [120]	randomized II/III Intensity-Modulated Proton Beam Therapy or Intensity-Modulated Photon Therapy	442	incidence of acute and chronic AEs OS PFS	QoL
CRUK/18/010 [121]	randomized III Intensity-Modulated Proton Beam Therapy or Intensity-Modulated Photon Therapy (TORPEdO trial)	180	UWPTCS feeding tube dependence	NTCP modelling

Abbreviations: RT—radiation therapy; CHT—chemotherapy; no.—number, PFS—progression-free survival; QoL—Quality of Life; LRF—locoregional failure; DF—distant failure; OS—overall survival; AE—adverse events; EFS—event-free survival; FACT-HN—functional assessment of cancer therapy—head and neck version; DMFS—distant metastases-free survival; DLT—dose-limiting toxicity; CRR—complete response rate; LRR—local/regional recurrence; DFS—disease-free survival; MDADI—M.D. Anderson Dysphagia Inventory; UWPTCS—University of Washington Physical Toxicity Composite Score; NTCP—normal tissue complication probability.

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
