# Peer review of "Human Papillomavirus-Related Non-Metastatic Oropharyngeal Carcinoma: Current Local Treatment Options and Future Perspectives"

_cancers, 2022, doi:10.3390/cancers14215385_

Round 1
Reviewer 1 Report
This is well written review article addressing most of the major issues in the management of HPV positive OPSCC. However, some issues can be further addressed
1. Line 481 -485 In the description of ORATOR 2 trial, as per the reported results (doi: 10.1001/jamaoncol.2022.0615.) both the treatment related deaths have happened in the TORS+neck dissection arm rather than RT arm. This needs to be corrected
2. Similarly Lines 239-242 : The reference to ECOG 3331 as a reference for comparison of TORS and low dose IMRT is wrong as E3331 trial did not compare TORS vs RT. The randomization was between two doses of adjuvant RT in the intermediate risk group
3. Line 566: The design of RTOG 106 and De-Escalate was not similar with RTOG 1016 being a non inferiority trial with OS endpoint and Deescalate being a superiority trial with toxicity endpoint.
4. Certain other risk factors (other than mentioned in the manuscript) have been used for risk stratification after TORS in recent randomized trials out of which perhaps the most important is negative but close margins (defined variably as 1-3 mm in ECOG 3331 and ORATOR 2, 1-5 mm in the PATHOS trial and 2 mm in ORATOR). I think the authors can shed some more light on this controversial issue in their review
Author Response
Dear Reviewer,
please find enclosed our responses to your comments regarding our manuscript
“Human papillomavirus-related non-metastatic oropharyngeal carcinoma: Current
local treatment options and future perspectives” by Michaela Svajdova et al.
R1: This is well-written review article addressing most of the major issues in the
management of HPV positive OPSCC. However, some issues can be further
addressed
Point 1. Line 481 -485 In the description of ORATOR 2 trial, as per the reported
results (doi: 10.1001/jamaoncol.2022.0615.) both the treatment related deaths have
happened in the TORS+neck dissection arm rather than RT arm. This needs to be
corrected.
Thank you for this comment. The information has been corrected (lines 510-513 in the
revised manuscript).
Point 2. Similarly Lines 239-242 : The reference to ECOG 3331 as a reference for
comparison of TORS and low dose IMRT is wrong as E3331 trial did not compare
TORS vs RT. The randomization was between two doses of adjuvant RT in the
intermediate risk group
Thank you for this comment. A correct information has been inserted into the revised
manuscript (lines 263-267).
Point 3. Line 566: The design of RTOG 106 and De-Escalate was not similar with
RTOG 1016 being a non-inferiority trial with OS endpoint and Deescalate being a
superiority trial with toxicity endpoint.
Thank you for this comment. The information has been added as well as corrected
(lines 590,594,601).
2/2
Bank account 87535621/0710
DM 7vqnyc6
VAT ID 00209805
E direct@mou.cz
T +420 543 131 111
F +420 543 211 169
Žlutý kopec 7, 656 53 Brno
Czech Republic
www.mou.cz
Point 4. Certain other risk factors (other than mentioned in the manuscript) have been
used for risk stratification after TORS in recent randomized trials out of which perhaps
the most important is negative but close margins (defined variably as 1-3 mm in ECOG
3331 and ORATOR 2, 1-5 mm in the PATHOS trial and 2 mm in ORATOR). I think the
authors can shed some more light on this controversial issue in their review.
Thank you. The valuable information regarding close surgical margins has been added
into the text as a new paragraph (lines 704-723).
We appreciate your time reading our revised manuscript in advance. Thank you for all
your valuable advice and comments.
Michaela Svajdova M.D., corresponding author on behalf of all co-authors
Clinic of Radiation Oncology
Masaryk Memorial Cancer Institute
Žlutý kopec 543/7, 602 00 Brno, Czech Republic
Telephone number: +421 911 618 265
E-mail address: michaela.svajdova@svetzdravia.com

Reviewer 2 Report
This is a detailed review of contemporary treatment, dilemmas and emerging therapeutic options in HPV+ carcinoma of the oropharynx. The text is rather well-written though (too) extensive, the references are appropriate and up-to-date.
1. What I miss is a table where the authors would briefly summarize dilemmas and new therapeutic options described in the text.
2. In the “Biology of HPV+ OPSCC” chapter, the indication of the % concordance of the results between the p16 IHS and HPV ISH/PCR tests is missing (how reliable is the p16 testing?)
3. The potential meaning of tumor-Infiltrating Lymphocytes (CD4+ and CD8+) is missing, e.g. Borsetto D et al. Cancers 2021;13:781.
4. When discussing early-stage cN0 disease, the role of modern imaging methods and techniques in establishing a reliable stage of the disease should be emphasized
5. p.7, L341-3: Incorrect reference - ref. 55 does not demonstrate that a total dose of cisplatin >200 mg/m2 is more effective than lower doses of cisplatin
Author Response
Dear Reviewer,
please find enclosed our responses to your comments regarding our manuscript “Human papillomavirus-related non-metastatic oropharyngeal carcinoma: Current local treatment options and future perspectives” by Michaela Svajdova et al.
R2: This is a detailed review of contemporary treatment, dilemmas and emerging therapeutic options in HPV+ carcinoma of the oropharynx. The text is rather well-written though (too) extensive; the references are appropriate and up-to-date.
Point 1. What I miss is a table where the authors would briefly summarize dilemmas and new therapeutic options described in the text.
Thank you very much for your suggestion. A new table (Table 3) has been added into the revised manuscript, listing all of the ongoing clinical trials mentioned in the text (lines 863-872).
Point 2. In the “Biology of HPV+ OPSCC” chapter, the indication of the % concordance of the results between the p16 IHS and HPV ISH/PCR tests is missing (how reliable is the p16 testing?)
Thank you for your comment. The information has been added into the manuscript (lines 87-91), a new reference has been added as well (ref. 18).
Point 3. The potential meaning of tumor-Infiltrating Lymphocytes (CD4+ and CD8+) is missing, e.g. Borsetto D et al. Cancers 2021;13:781.
Thank you very much. The information has been added (lines 143-147, ref. 25).
Point 4. When discussing early-stage cN0 disease, the role of modern imaging methods and techniques in establishing a reliable stage of the disease should be emphasized.
Thank you. The required information has been added into the text (lines 172-182, ref.30).
Point 5. P.7, L341-3: Incorrect reference - ref. 55 does not demonstrate that a total dose of cisplatin >200 mg/m2 is more effective than lower doses of cisplatin.
Thank you very much. The information has been corrected and a new reference has been added into the manuscript (lines , ref. 368-371, ref. 58).
We appreciate your time reading our revised manuscript in advance. Thank you for all your valuable advice and comments.
Michaela Svajdova M.D., corresponding author on behalf of all co-authors
Clinic of Radiation Oncology
Masaryk Memorial Cancer Institute
Žlutý kopec 543/7, 602 00 Brno, Czech Republic
Telephone number: +421 911 618 265
E-mail address: michaela.svajdova@svetzdravia.com
